# High Incidence of False Positives in *EGFR* S768I Mutation Detection Using the Idylla qPCR System in Non-Small Cell Lung Cancer Patients

**DOI:** 10.3390/diagnostics15030321

**Published:** 2025-01-30

**Authors:** Miguel Carnero-Gregorio, Enzo Perera-Gordo, Vanesa de-la-Peña-Castro, Jesús María González-Martín, Julio José Delgado-Sánchez, Carmen Rodríguez-Cerdeira

**Affiliations:** 1Department of Pathological Anatomy, Hospital Universitario de Gran Canaria Dr. Negrín, Barranco de la Ballena, s/n, 35010 Las Palmas de Gran Canaria, Spain; epergor@gobiernodecanarias.org (E.P.-G.); jdelsanr@gobiernodecanarias.org (J.J.D.-S.); 2Fundación Vithas, Grupo Hospitalario Vithas, 28043 Madrid, Spain; 3Fundación Canaria Instituto de Investigación Sanitaria de Canarias (FIISC), Barranco de la Ballena, s/n, Edf. Anexo al Hospital Universitario de Gran Canaria Dr. Negrín, 35019 Las Palmas de Gran Canaria, Spain; nesa206@gmail.com; 4Research Unit, Hospital Universitario de Gran Canaria Dr. Negrín, C/Barranco de la Ballena, s/n, 35010 Las Palmas de Gran Canaria, Spain; josu.estadistica@gmail.com; 5Dermatology Department, Grupo Hospitalario (CMQ Concheiro), Manuel Olivié 11, 36203 Vigo, Spain; 6Department of Health Sciences, University of Vigo, Campus of Vigo, As Lagoas, 36310 Vigo, Spain

**Keywords:** molecular diagnostics, non-small cell lung cancer, EGFR mutation testing, S768I variant, false positives, qPCR, next-generation sequencing

## Abstract

**Background/Objectives:** The accurate detection of *EGFR* mutations, particularly the rare S768I variant, is crucial for guiding treatment decisions in non-small cell lung cancer (NSCLC) patients. This study investigated the incidence of false positives in S768I mutation detection using the Idylla^TM^ qPCR system and compared results with next-generation sequencing (NGS). **Methods:** A prospective observational study was conducted at the Dr. Negrín University Hospital between July 2023 and July 2024. Six NSCLC patient samples with S768I variant detection by Idylla^TM^ were analyzed from all NSCLC cases tested during the study period. Initial testing was performed on tissue samples (Idylla1), followed by replicate analysis using extracted DNA (Idylla2). Results were compared with NGS as the reference method. Statistical analysis included the calculation of sensitivity, specificity, accuracy, and Kappa concordance index. **Results:** Initial Idylla testing showed an 80% false positive rate, with only one of five positive results confirmed by NGS. The first analysis demonstrated high sensitivity (100%) but low specificity (20%), with an accuracy of 0.333 and poor concordance with NGS (Kappa = 0.077). Repeat testing using extracted DNA showed improved performance, with increased accuracy (0.833) and better agreement with NGS (Kappa = 0.571). Analysis of amplification curves revealed that false positives typically showed normalized fluorescence values below 12 points, with no clear correlation between false positives and factors such as sample quantity or tumor content. **Conclusions:** While the Idylla^TM^ system shows high sensitivity for S768I detection, its initial specificity is problematic, leading to frequent false positives. These findings emphasize the importance of confirming positive S768I results through alternative methods like NGS, particularly when these results could influence therapeutic decisions. Results suggest the need to refine the system’s interpretation algorithms to improve specificity.

## 1. Introduction

In 2020, 19.3 million new cancer cases were diagnosed worldwide, with 10 million cancer-related deaths recorded. Lung cancer (LC) is the leading cause of cancer death, with an incidence of 2.5 million cases and approximately 1.8 million deaths per year, ranking first in both incidence and mortality and accounting for 18.4% of all cancer deaths globally [1,2,3]. LC is divided into two major groups: non-small cell lung cancer (NSCLC), which is the most common and represents approximately four out of five cases, and small cell lung cancer (SCLC), which accounts for approximately 15% of cases [4,5,6].

In the last two decades, advances in understanding the molecular biology of NSCLC have enabled the identification of genetic alterations that play a crucial role in disease pathogenesis and progression [7]. Some of these variants are found in the *EGFR* gene, which codes for the epidermal growth factor receptor, and have emerged as one of the most significant findings, with direct implications for patient therapeutic management [7,8,9]. The *EGFR* gene is altered in approximately 10% to 15% of NSCLC patients in Western populations and up to 40% to 50% in Asian populations [9,10]. These alterations result in the constitutive activation of the *EGFR* receptor, promoting cell proliferation and tumor survival [9]. Their identification has revolutionized NSCLC treatment and is key to its personalization, as patients with *EGFR*-activating variants show a greater response to tyrosine kinase inhibitors (TKIs) compared to conventional chemotherapy [10].

The S768I variant, found in exon 20 of the EGFR gene, is one of the less common but clinically significant alterations, representing approximately 1% of all mutations occurring in this gene [11]. This variant involves the substitution of a serine residue (polar amino acid) with an isoleucine residue (nonpolar amino acid) at codon 768 of the protein encoded by this gene and has been associated with both sensitivity and resistance to different TKIs [12,13,14]. Despite its lower frequency compared to other mutations, such as L858R or exon 19 deletions, the detection of S768I is equally important due to its specific therapeutic implications [12,15].

The identification of this variant is commonly performed using real-time polymerase chain reaction (qPCR) techniques or through next-generation sequencing (NGS). One of the qPCR-based tests used in healthcare centers is the Idylla™ *EGFR* Mutation Test (CE-IVD) (Biocartis, Mechelen, Belgium). This system performs all necessary steps automatically, reducing errors, handling time, and risk of cross-contamination, allowing for a rapid and reproducible evaluation of *EGFR* variants, comparable to other techniques, offering high sensitivity and specificity [10,16,17]. However, the Idylla™ *EGFR* Mutation Test based on qPCR is not free from errors, and cases of false negatives or false positives can occur.

Given the importance of precision in molecular diagnosis to guide treatment decisions in NSCLC patients, it is necessary to investigate and understand the underlying causes of false positives and negatives for the S768I mutation in the Idylla^TM^ system. This will help improve diagnostic accuracy and contribute to optimizing the use of this platform in clinical practice.

The objective of this article is to explore the prevalence of false positives in the detection of the *EGFR* gene S768I mutation after one year of using the Biocartis qPCR Idylla^TM^ platform at the Dr. Negrín University Hospital of Gran Canaria (Gran Canaria, Spain), analyzing the possible causes contributing to these incorrect results and discussing possible solutions.

## 2. Materials and Methods

### 2.1. Study Design and Patient Selection

This prospective observational study was conducted between 1 July 2023 and 1 July 2024 in the Department of Pathological Anatomy at the Dr. Negrín University Hospital of Gran Canaria (HUGCDN) in Las Palmas de Gran Canaria, Spain. Patients with non-small cell lung cancer (NSCLC) diagnosed in this department were included, regardless of their pathological stage. During the evaluation period, all NSCLC cases diagnosed in our Pathology Department underwent comprehensive molecular testing as part of routine clinical practice, with each sample analyzed in parallel using both the Idylla^TM^ system and NGS for a complete panel of actionable mutations, including *EGFR*, *KRAS*, *ALK*, *ROS1*, *MET*, and *RET*. This parallel testing strategy was implemented as part of an ongoing departmental project.

### 2.2. Sample Collection and Processing

Previously extracted paraffin-embedded (FFPE) biopsy samples were used. The tumor area and tumor content of the samples were determined by pathologists from the Department of Pathological Anatomy. *EGFR* gene variant determination was performed using two methods: qPCR and next-generation sequencing (NGS).

### 2.3. Idylla^TM^ System Analysis

The determination of pathogenic variants in *EGFR* by qPCR was performed using the Idylla^TM^ *EGFR* Mutation Test cartridge (Biocartis, Mechelen, Belgium) on the Idylla^TM^ Platform. This fully automated system integrates multiple steps of molecular testing into a single cartridge-based process. The technology combines automated sample preparation and mutation detection through three key steps: tissue processing using a combination of chemical reagents, enzymes, heat, and high-intensity focused ultrasound; multiplexed PCR amplification with allele-specific primers; and real-time detection using fluorescence-labeled probes. The system can detect 51 *EGFR* mutations in exons 18, 19, 20, and 21, offering a comprehensive coverage of clinically relevant variants.

The procedure was carried out according to the manufacturer’s instructions [18], with a total processing time of approximately 2.5 h and an actual handling time of less than 2 min. Briefly, one or more tissue sections totaling between 5 and 10 µm in thickness, all with a minimum of 20% tumor cells, were loaded into the Idylla^TM^ *EGFR* Mutation Test cartridges. The cartridges were inserted into the Idylla^TM^ instrument, which automatically performed DNA extraction, real-time PCR, and results analysis. Deparaffinization, tissue disruption, and cell lysis are achieved through a combination of chemical reagents, enzymes, heat, and high-intensity focused ultrasound. Real-time PCR uses allele-specific primers and fluorescence-labeled probes for variant detection [19]. For DNA analyses extracted via Idylla^TM^, the recommendations of Bocciarelli et al. (2020) were followed, where they recommend that the minimum quantity should be greater than 25 ng of DNA [20]. Results are shown directly on the Idylla^TM^ console in report form and classified as positive, negative, or invalid. Each run included internal control checks for sample processing and amplification to ensure result validity. Amplification curves and Cq values were reviewed for all samples. This was performed by connecting to the Idylla^TM^ Explore web application (https://idyllaexplore.biocartis.com/ (accessed on 1 July 2023)), where these results can be viewed and curves analyzed. Another parameter to consider is the ΔCq, which is the difference between the control sample Cq values and the S768I variant Cq values. Lower ΔCq values typically indicate a higher abundance of the target mutation, while higher values suggest a lower abundance or potential false positive signals. According to the manufacturer’s specifications, the Limit of Detection (LOD) for the S768I mutation varies depending on the total *EGFR* Cq value. With 1000 input copies (Total *EGFR* Cq = 21.3), the LOD is 2.5%, while with 2500 copies (Total *EGFR* Cq = 19.8), it improves to 1.4%. Results should be interpreted considering these analytical sensitivity thresholds.

### 2.4. Next-Generation Sequencing Analysis

For NGS analysis, DNA was extracted from the same tissue samples. Nucleic acids were extracted and purified using the RecoverAll™ Total Nucleic Acid Isolation Kit for FFPE (Invitrogen™, ThermoFisher Scientific, Waltham, MA, USA), according to manufacturer specifications and guidelines and quantified using the Qubit™ 4 Fluorometer (Invitrogen™, San Diego, CA, USA). Prior to NGS analysis, tumor cell content was assessed for each sample by a pathological review of adjacent sections. The same tumor area evaluated for Idylla^TM^ testing was used for DNA extraction for NGS to ensure comparable tumor content between both methods. All samples met the minimum requirement of 20% tumor cellularity for reliable variant detection. Library preparation, sequencing, and data analysis were carried out using the Ion GeneStudio™ S5 System platform (ThermoFisher^TM^ Scientific Inc.) and the Oncomine™ Focus Assay panel (ThermoFisher^TM^ Scientific Inc.). Only one sample was sequenced and analyzed using the Ion Torrent™ Genexus™ System platform (ThermoFisher^TM^ Scientific Inc.) and the Oncomine™ Focus Assay panel (ThermoFisher^TM^ Scientific Inc.) due to equipment updates performed in our laboratory in January 2024. In this case, library preparation was conducted using the Ion Torrent™ Genexus™ System itself in an automated manner. Both panels have coverage for *EGFR* exons 18, 19, 20, and 21, including the S768I variant [21,22].

For all samples, a minimum coverage depth of 1000 reads at the S768I locus was required for reliable variant calling. The mean coverage achieved across samples at this specific locus was approximately 2000 reads, ensuring robust mutation detection capability.

### 2.5. Statistical Analysis

Statistical analysis was performed to evaluate the concordance between results obtained by Idylla^TM^ and NGS in detecting the *EGFR* S768I variant. Two Idylla datasets were used, corresponding to the first analysis (Idylla1) and the duplicate (Idylla2), to compare with the results obtained by NGS as the reference method. To evaluate the concordance between the two variables, the following statistics were calculated: Kappa index, accuracy, prevalence, sensitivity, specificity, positive predictive value, and negative predictive value. The ROC curve was calculated between a quantitative variable and a dichotomous one (NGS) to obtain the optimal cutoff point to discriminate between positive and negative. The statistical program used was R Core Team 2024, version 4.3.3 (https://www.r-project.org/ (accessed on 1 September 2024)).

## 3. Results

### 3.1. Patient Characteristics

Among all *EGFR* mutations detected, the distribution of variant types was consistent with published frequencies, with the S768I variant representing a small fraction of cases, in line with its reported frequency of approximately 1% of all *EGFR* mutations.

From all cases analyzed, six patients were identified in whom Idylla^TM^ detected the S768I variant of the *EGFR* gene. These cases were further analyzed using the Idylla^TM^ system in duplicate (Idylla1 and Idylla2) and NGS for detailed evaluation. Among these six patients, the mean age was 67.5 ± 6.08 years, with five males (M) and one female (F). Tumor histological types included large cell neuroendocrine carcinoma (*n* = 1), adenocarcinoma (*n* = 2), squamous cell carcinoma (*n* = 2), and unspecified lung neoplasm (*n* = 1). All patients were metastatic (stage IVa or IVb) (Table 1).

### 3.2. Performance of the Idylla^TM^ System Detection

As part of our routine molecular diagnostic workflow, all *EGFR* mutation results from Idylla^TM^ testing were cross-validated by NGS analysis. Prior to this systematic evaluation, our laboratory had no experience with S768I variant detection using the Idylla^TM^ system, as this technique was incorporated into our lab routine in June 2023. The concordance between Idylla^TM^ and NGS for other common *EGFR* mutations (such as exon 19 deletions and L858R) was high, with discrepancies primarily observed in S768I variant detection.

The Idylla^TM^ system identified a total of seven S768I alterations (seven in 2023 and one in 2024) in six samples analyzed in duplicate: five in the first analysis (Idylla1) and two in the replicates (Idylla2). Tissue samples (biopsies or cell blocks) were analyzed using the Idylla^TM^ system in duplicate (Idylla1 and Idylla2). For Idylla2 replicates, the same extracted DNA that was used for NGS sequencing was used due to either a lack of additional tissue sample or insufficient tumor representation in the remaining block (except for sample 107, where more material was available) (Table 2).

The Cq values for total *EGFR* (control) were relatively consistent between samples, being 24.6 ± 3.02 (range: 21.94–30.44) for Idylla1 and 22.46 ± 1.27 (range: 21.3–24.9) for Idylla2, suggesting similar DNA quantity in all samples and generally good for analysis, with only one value far from the mean (Sample 142-Cq 30.44). According to established LOD parameters, optimal detection sensitivity (2.5%) requires a minimum total *EGFR* Cq of 21.3. While some of our samples met this threshold, others showed higher Cq values, potentially affecting the reliability of mutation detection. The ΔCq values for samples called positive for S768I ranged from 5.24 to 9.71, with the confirmed true positive case (sample 107) showing the lowest ΔCq values.

During the study period, S768I amplification curves were observed in samples ultimately classified as negative. In these cases, while some amplification was detected, the curves typically showed normalized fluorescence values below 12 points. This pattern was consistently observed across negative samples, suggesting it may represent background amplification rather than true mutation detection. The amplification curves obtained by the Idylla^TM^ system for the detection of the *EGFR* gene S768I variant were analyzed in all samples. Figure 1 shows the results for the six samples analyzed.

In all samples, except for 142, the amplification of the S768I curve was observed when tissue was used, although with variations in its profile and Cq values. When repeating the test using extracted DNA, all gave negative results, except for sample 142. Sample 107 was repeated with a biopsy sample, as more material was available for analysis.

Although tumor cell content varied between samples (20% to 80%), there was no clear correlation between tumor cellularity and the occurrence of false positive results. Both high tumor content samples (e.g., sample 143 with 80%) and lower tumor content samples (e.g., sample 195 with 20%) showed discrepant results between Idylla and NGS, suggesting that variations in tumor content alone do not explain the observed discrepancies.

### 3.3. Comparison with NGS Results

Concerning NGS sequencing, all samples were analyzed using the Ion GeneStudio™ S5 System sequencer and the OFA panel, except for sample 195, which was analyzed using the Ion Torrent™ Genexus™ System platform and the OPA panel. The DNA concentration of the samples for NGS use ranged between 1.02 ng/μL and 19.4 ng/μL. NGS detected only one S768I alteration in the same analyzed samples (Table 3).

To better visualize the concordance between methods, a direct comparison of results obtained by both Idylla runs and NGS is presented in Table 4.

A low ΔCq could suggest a higher relative abundance of the S768I variant, although this should be interpreted with caution given that most of these detections turned out to be false positives in the NGS validation. The only sample in which both Idylla^TM^ and NGS detected the S768I variant corresponded to sample 132.

### 3.4. Statistical Analysis and False Positive Rates

The statistical analysis performed on the results obtained by Idylla^TM^ and NGS in detecting the *EGFR* S768I variant revealed significant findings. For Idylla1, of the six samples in which the S768I variant was detected, only one was confirmed by NGS. This suggests a false positive rate of 80% (4/5) for S768I detection by Idylla^TM^ in our case series. The accuracy was 0.333 (95% CI: 0.04–0.78). The Kappa concordance index was 0.077, indicating very weak concordance with NGS, suggesting a substantial discrepancy between both methods.

The high sensitivity (100%; 95% CI: 0.03–1) but low specificity (20%; 95% CI: 0.01–0.72) of Idylla1 implies that the system effectively detects all positive cases but also produces a considerable number of false positives. This is reflected in the low positive predictive value (PPV), which was 0.2 (95% CI: 0.01–0.72), indicating that only 20% of Idylla1 positive results are true positives, according to NGS. The high negative predictive value (NPV), which was 1 (95% CI: 0.03–1), suggests that Idylla1’s negative results are reliable, although these data should be interpreted with caution, given the small sample size. The ROC curve analysis for Idylla1 (Figure 2A), with an area under the curve (AUC) of 0.8 and an optimal cutoff point of 23.345, indicates good overall test performance but also suggests that adjusting the positivity threshold could improve its specificity.

For Idylla2, a notable improvement in diagnostic performance was observed. The increase in the Kappa index to 0.571 indicates moderate agreement with NGS, representing a substantial improvement compared to Idylla1. Accuracy increased from 0.333 to 0.833 (95% CI: 0.36–1), and specificity improved to 80% (95% CI: 0.28–0.99) while maintaining 100% sensitivity (95% CI: 0.03–1). The PPV increased to 0.5 (95% CI: 0.01–0.99), while the NPV remained at 1 (95% CI: 0.4–1). The optimal cutoff point for Idylla2 was 22.38, with an AUC of 0.8 and identical sensitivity, specificity, and accuracy values to those of Idylla1 (Figure 2B). This suggests that the modifications introduced in Idylla2 (performing the test with extracted DNA instead of tissue) succeeded in significantly reducing false positives without compromising the detection of true positive cases. The increase in positive predictive value to 0.5 indicates that, in this case, half of Idylla2’s positive results are confirmed by NGS, representing a considerable improvement in the reliability of positive results.

The ROC curve analysis yielded an AUC of 0.8 for both Idylla1 and Idylla2, indicating good discriminatory ability. This value suggests that the test can effectively distinguish between true positive and negative cases 80% of the time. However, the clinical implications of this performance metric should be considered carefully, particularly given our small sample size. The identical AUC values between Idylla1 and Idylla2, despite their different specificity profiles, suggest that while both versions have the similar overall diagnostic capability, the improved specificity in Idylla2 makes it more suitable for clinical implementation.

The descriptive statistics revealed that the control Cq values for Idylla1 had a mean of 24.6 (±3.02), with a range from 21.94 to 30.44, while for Idylla2, the mean was 22.46 (±1.27), with a range from 21.3 to 24.9. This reduction in the variability of Cq values in Idylla2 could partially explain the improvement in diagnostic performance. It is important to note that due to the small sample size (n = 6), the confidence intervals are wide, which limits the precision of the estimates and underscores the need to interpret these results with caution.

## 4. Discussion

Alterations occurring in the *EGFR* gene, especially those in exons 18 to 21, are determinants for response to tyrosine kinase inhibitors (TKIs), a class of targeted therapies that have significantly improved prognosis in NSCLC patients [9,10,23,24]. The S768I mutation, although less common, has important clinical implications [12,25], which highlights the need for accurate detection to select appropriate treatment. However, a precise detection of these mutations is critical, as false positives can lead to incorrect therapeutic decisions and suboptimal patient management [26,27].

qPCR is a widely used technique for mutation detection due to its high sensitivity and specificity. Nevertheless, each qPCR platform and method can have different levels of precision and propensity for errors [28]. The Biocartis Idylla^TM^ system is an automated molecular diagnostic instrument designed to facilitate the rapid and accurate detection of genetic mutations, including those in *EGFR*. This system is valued for its ability to provide quick and reliable results, allowing real-time results without the need for complex sample preparation, which is particularly useful in clinical settings where time is a critical factor [29,30]. The Idylla^TM^ system, while offering significant advantages in terms of automation and rapid turnaround time, has several important limitations that need to be considered. The reliability and validity of results depend heavily on sample quality and preparation, with multiple preanalytical factors potentially affecting performance. These include suboptimal sample collection, handling procedures, variations in tissue fixation methods, and the use of stained tissues. Additionally, paraffin samples with high melting temperatures may interfere with proper DNA extraction. The system also has specific requirements for minimum tumor content and DNA quantity that must be met for reliable results, as evidenced by our findings about Cq values and sample adequacy. These technical limitations, combined with the observed specificity issues for S768I detection in our study, suggest that the system’s performance should be carefully monitored and validated. The high prevalence of false positives observed specifically in the S768I detection indicates that a more cautious approach is necessary, including mandatory cross-validation with other detection methods such as NGS. Given these limitations, we recommend against using the Idylla^TM^ system as a sole detection method for the S768I mutation, particularly when these results could influence critical therapeutic decisions.

Some studies have reported cases of false positives in S768I mutation detection using the Idylla platform. In a multicenter study evaluating Idylla’s performance, a false positive for S768I was observed that was not confirmed by reference methods [29]. Similarly, another study analyzing decalcified bone metastasis samples also reported a false positive for this mutation [31]. The occurrence of these false positives raises important concerns. First, it can lead to an overestimation of this mutation’s prevalence in the NSCLC patient population. More critically, it can result in inappropriate therapeutic decisions, potentially exposing patients to unnecessary or ineffective treatments. False positives (and false negatives) in S768I detection can arise for various reasons, including nonspecific amplification, cross-contamination, and technical problems inherent to the assay design [26,32,33]. The incidence of false positives not only compromises diagnostic accuracy but can also lead to erroneous therapeutic decisions, such as an inappropriate administration of TKIs, which can result in adverse effects and treatment resistance [11].

The Idylla^TM^ System showed a high sensitivity for S768I signal detection, as evidenced by amplification in all samples. However, the low specificity (high false positive rate) suggests that signal detection does not always correspond to the actual presence of the variant. It is important to note that all S768I curves crossed the positivity threshold established by the Idylla^TM^ System, which led to the classification of these samples as positive for the variant. This highlights the need to reevaluate the positivity criteria to reduce false positives.

Regarding the possible causes of the false positives found, there was no clear relationship between the amount of sample used or the *EGFR* control Cq value and the likelihood of obtaining a false positive. For example, patient 195, with the lowest Cq (21.8), indicative of good sample quality, also turned out to be a false positive. Additionally, false positives were observed in various histological types, including adenocarcinoma, squamous cell carcinoma, and large cell neuroendocrine carcinoma, suggesting that tumor type would not be a determining factor in the appearance of false positives. As for tumor cell content, it varied widely between samples (from 25% to 80%), with no apparent correlation with the probability of false positives.

The S768I curves showed different patterns of exponential growth and plateau between samples, reflecting differences in amplification efficiency and/or the amount of variant present. It was observed that amplification curves for S768I, in cases of false positives, tended to show a normalized fluorescence of fewer than 12 points, which could be a parameter to consider for determining false positives. Notably, this high rate of false positives appears to be specific to the S768I variant, as concordance between Idylla^TM^ and NGS was substantially better for other *EGFR* mutations tested in our laboratory in the study period. This suggests that the technical challenges in accurate detection may be particularly relevant to this specific variant. The presence of low-level amplification curves in confirmed negative samples suggests that the S768I detection may be particularly susceptible to background noise. This observation, combined with our findings about false positives, indicates that a careful evaluation of amplification patterns and fluorescence thresholds is crucial for accurate mutation calling. A study with more samples would be necessary to determine if this is a value that can be established as a cutoff point.

Taken together, these results suggest that although Idylla^TM^ shows high sensitivity in detecting the S768I variant, its initial specificity (Idylla1) was problematic, leading to a high false positive rate. The improvements observed in Idylla2 are promising, indicating that with appropriate adjustments, the system could achieve more balanced and reliable diagnostic performance using extracted DNA from samples. However, given the persistent discrepancy with NGS, especially in the first analysis (Idylla 1), these findings highlight the importance of confirming positive *EGFR* S768I results obtained by Idylla^TM^ through alternative methods such as NGS, especially in cases where the detection of this variant could influence critical therapeutic decisions.

It is also important to consider the economic and resource implications of false positives in diagnostic testing. Errors in molecular diagnosis can lead to unnecessary treatments, additional follow-ups, and potential delays in the administration of more appropriate therapies. This not only affects the quality of patient care but also has implications for efficiency and resource allocation in healthcare systems [30].

To further validate these findings, multicenter studies would be valuable. Such studies could determine whether the high false positive rates for S768I detection are consistently observed across different laboratories using the Idylla^TM^ platform or if they are influenced by local factors such as sample processing variations or environmental conditions.

## 5. Conclusions

In conclusion, although the Idylla^TM^ system demonstrates high sensitivity for detecting signals associated with S768I, the high rate of false positives observed in this study underscores the need to interpret these results with caution. Confirmation by alternative methods, such as NGS, is recommended, especially in cases where the detection of this variant could influence important therapeutic decisions. Furthermore, these findings suggest the need to refine the Idylla^TM^ system’s interpretation algorithms to improve its specificity in detecting the S768I variant of the *EGFR* gene in NSCLC samples.

## Figures and Tables

**Figure 1 diagnostics-15-00321-f001:**
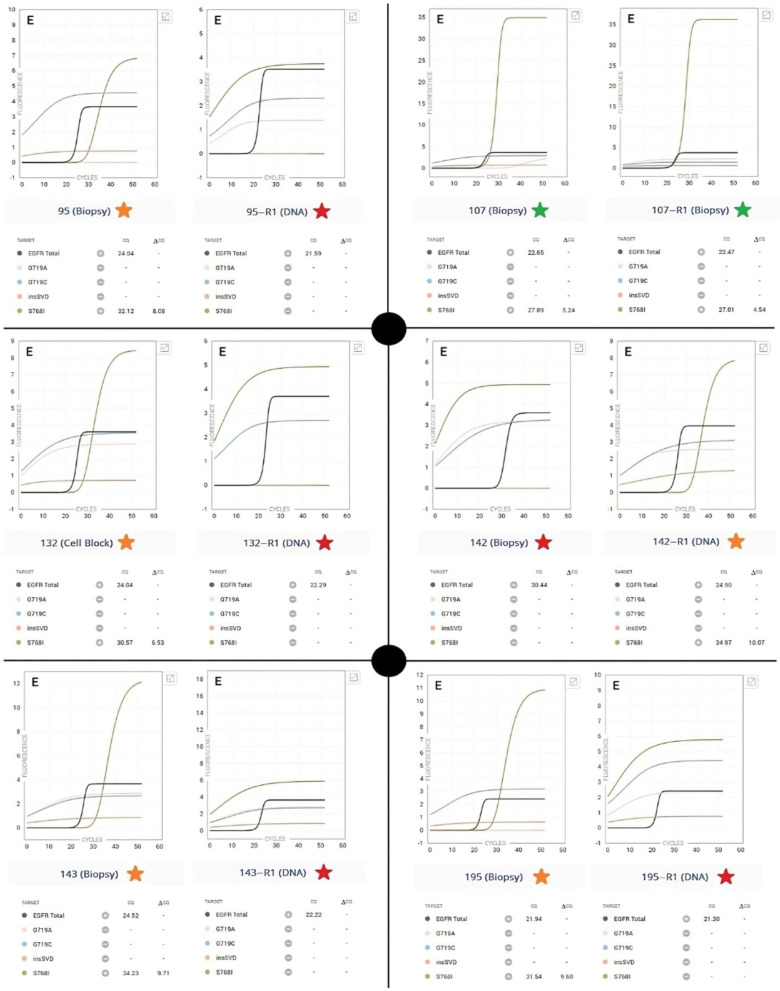
Amplification curves for EGFR S768I variant detection in six samples analyzed by Idylla1 and Idylla2 (R1). Curves show total EGFR (control), G719A, G719C, insSVD, and S768I amplification. Green stars are true positives, orange stars are false positives, and red stars are negative results. Sample types are indicated in parentheses.

**Figure 2 diagnostics-15-00321-f002:**
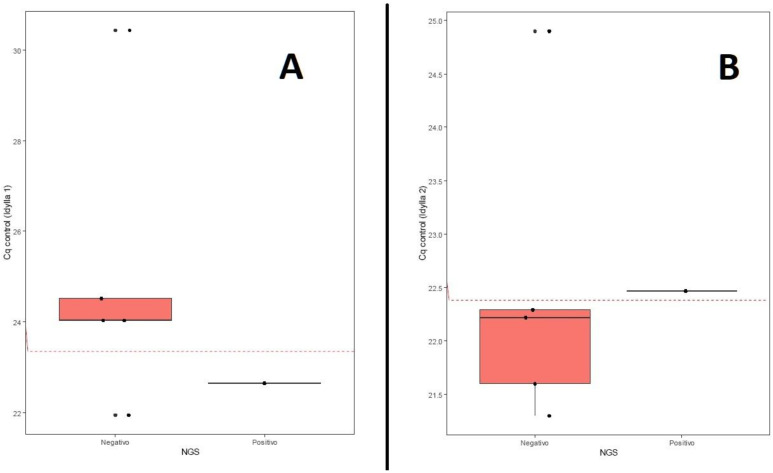
ROC curve analysis comparing Idylla^TM^ and NGS results. (**A**) ROC curve for Idylla1 (AUC = 0.8, optimal cutoff = 23.345). (**B**) ROC curve for Idylla2 (AUC = 0.8, optimal cutoff = 22.38).

**Table 1 diagnostics-15-00321-t001:** Clinical and pathological characteristics of patients with S768I detected by Idylla^TM^.

Sample	Age	Gender	Tumor Area	% Tumor Cells	Tumor Type	Stage
95	75	M	17 mm^2^	60%	High-grade neuroendocrine tumor, suspicious for large cell neuroendocrine carcinoma	T4N2M1b (IVA)
107	72	M	24 mm^2^	80%	NSCLC, most likely adenocarcinoma	T4N2M1a (IVA)
132	59	F	4 mm^2^	65%	NSCLC, compatible with adenocarcinoma	T4N3M1c (IVB)
142	73	M	4 mm^2^	25%	Lung neoplasm	Unknown (IVA)
143	64	M	15 mm^2^	80%	NSCLC, most likely squamous cell carcinoma	T4N0M1a (IVA)
195	62	M	44 mm^2^	20%	NSCLC, most likely squamous cell carcinoma	T3N3M1c (IVB)

**Table 2 diagnostics-15-00321-t002:** Results of the Idylla^TM^ analysis.

Sample	Sample Type	Quantity	Result	Cq (Total *EGFR*)	Cq (S768I)	ΔCq
95	Biopsy	2 slides (3 µm each)	Positive	24.04	32.12	8.08
95-R1	Extracted DNA	100 ng	Negative	21.59	-	-
107	Biopsy	1 slide (5 µm)	Positive	22.65	27.89	5.24
107-R1	Biopsy	1 slide (5 µm)	Positive	22.47	27.01	4.54
132	Cell block	2 slides (3 µm each)	Positive	24.04	30.57	6.53
132-R1	Extracted DNA	126 ng	Negative	22.29	-	-
142	Biopsy	1 slide (5 µm)	Negative	30.44	-	-
142-R1	Extracted DNA	40.8 ng	Positive	24.90	34.97	10.07
143	Biopsy	2 slides (5 µm each)	Positive	24.52	34.23	9.71
143-R1	Extracted DNA	155.2 ng	Negative	22.22	-	-
195	Biopsy	2 slides (3 µm each)	Positive	21.94	31.54	9.60
195-R1	Extracted DNA	101.16 ng	Negative	21.30	-	-

**Table 3 diagnostics-15-00321-t003:** NGS analysis parameters and results. Sample 195 shows higher read depth due to analysis on the Ion Torrent™ Genexus™ System platform versus the Ion GeneStudio™ S5 System used for other samples.

Sample	[DNA]	Read Depth	Result
95	3.18 ng/µL	1995	Negative
107	9.40 ng/µL	1944	Positive
132	4.20 ng/µL	1995	Negative
142	1.02 ng/µL	1993	Negative
143	19.4 ng/µL	1996	Negative
195	5.62 ng/µL	4739	Negative

**Table 4 diagnostics-15-00321-t004:** Comparison of results between detection methods for each sample analyzed.

Sample	Sample Type	Idylla1 Result	Idylla2 Result	NGS Result	Method Concordance
95	Biopsy	Positive	Negative	Negative	Idylla2 = NGS
107	Biopsy	Positive	Positive	Positive	Both = NGS
132	Cell Block	Positive	Negative	Negative	Idylla2 = NGS
142	Biopsy	Negative	Positive	Negative	Idylla1 = NGS
143	Biopsy	Positive	Negative	Negative	Idylla2 = NGS
195	Biopsy	Positive	Negative	Negative	Idylla2 = NGS

## Data Availability

The original contributions presented in this study are included in the article. Further inquiries can be directed to the corresponding author(s).

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
