# Peer review of "High Incidence of False Positives in EGFR S768I Mutation Detection Using the Idylla qPCR System in Non-Small Cell Lung Cancer Patients"

_diagnostics, 2025, doi:10.3390/diagnostics15030321_

Round 1
Reviewer 1 Report
Comments and Suggestions for Authors
Comments to the Author
The manuscript titled "Increased False Positives in Detecting EGFR S768I Mutation Using the Idylla RT-PCR System in Non-Small Cell Lung Cancer Patients" investigates the detection of EGFR mutations, particularly the S768I variant, which can significantly influence treatment decisions in non-small cell lung cancer (NSCLC) patients. The findings highlight the importance of validating S768I-positive results through alternative methods such as NGS. However, several points should be addressed:
1-In the Materials and Methods section, all elements are currently grouped under a single heading. It is recommended to separate them into distinct subsections, for example: Study Cases, The Idylla™ System, and The Idylla™ EGFR S768I Mutation Test. This structure will enhance clarity and improve the organization of the section. Similarly, for the Results section, it is advisable to structure the content under separate subsections. This could include headings such as Performance of the Idylla™ System, False Positive Rates for the EGFR S768I Mutation, and Comparison with NGS Results. This approach will improve readability and allow readers to easily navigate and interpret the findings.Furthermore, in the Materials and Methods section, it is important to provide a detailed explanation of the Idylla™ system, including its principles, workflow, and how it is used to detect EGFR mutations. This additional detail will offer readers a better understanding of the methodology and the technology employed in the study.
2-Including comparative data between the Idylla RT-PCR and NGS methods would strengthen the conclusions.
3-Your analysis of ROC curves provides valuable diagnostic insights. It might enhance understanding if you briefly explain what the AUC values imply for clinical applicability.
4-Put a reference for lines 27,280,282,288
5-It is recommended to provide a more comprehensive discussion of the limitations associated with the use of the Idylla RT-PCR system
6-The phrase "between July 1, 2023, and July 1, 2024" is mentioned in both the Materials and Methods and Results sections. However, it is unnecessary to repeat this information in the Results section, as it has already been detailed in the Materials and Methods section.
Author Response
Reviewer 1
The manuscript titled "Increased False Positives in Detecting EGFR S768I Mutation Using the Idylla RT-PCR System in Non-Small Cell Lung Cancer Patients" investigates the detection of EGFR mutations, particularly the S768I variant, which can significantly influence treatment decisions in non-small cell lung cancer (NSCLC) patients. The findings highlight the importance of validating S768I-positive results through alternative methods such as NGS. However, several points should be addressed:
- In the Materials and Methods section, all elements are currently grouped under a single heading. It is recommended to separate them into distinct subsections, for example: Study Cases,The Idylla™ System, and The Idylla™ EGFR S768I Mutation Test. This structure will enhance clarity and improve the organization of the section. Similarly, for the Results section, it is advisable to structure the content under separate subsections. This could include headings such as Performance of the Idylla™ System, False Positive Rates for the EGFR S768I Mutation, and Comparison with NGS Results. This approach will improve readability and allow readers to easily navigate and interpret the findings. Furthermore, in the Materials and Methods section, it is important to provide a detailed explanation of the Idylla™ system, including its principles, workflow, and how it is used to detect EGFR mutations. This additional detail will offer readers a better understanding of the methodology and the technology employed in the study.
We have restructured both sections with clear subsections to enhance readability. The Materials and Methods now includes separate subsections for study design, sample processing, Idylla™ analysis, NGS analysis, and statistical methods. Similarly, the Results section has been reorganized to present our findings in a more systematic way.
We have enhanced the description of the Idylla™ system in the Materials and Methods section to provide a more comprehensive explanation of its principles, integrated workflow, and technological capabilities.
- Including comparative data between the Idylla RT-PCR and NGS methods would strengthen the conclusions.
We have added a new comprehensive table showing direct result comparisons.
- Your analysis of ROC curves provides valuable diagnostic insights. It might enhance understanding if you briefly explain what the AUC values imply for clinical applicability.
We have added a detailed explanation in the Results section that contextualizes the AUC value of 0.8 and its practical significance for diagnostic decision-making. This addition helps readers better understand how the statistical performance metrics translate to clinical utility.
- Put a reference for lines 27, 280, 282, 288
Corrected
- It is recommended to provide a more comprehensive discussion of the limitations associated with the use of the Idylla RT-PCR system
We have expanded and consolidated our discussion of the Idylla system's limitations into a comprehensive paragraph that addresses both technical and methodological constraints.
- The phrase "between July 1, 2023, and July 1, 2024" is mentioned in both the Materials and Methods and Results However, it is unnecessary to repeat this information in the Results section, as it has already been detailed in the Materials and Methods section.
We have removed the redundant mention of the study period from the Results section.
Reviewer 2 Report
Comments and Suggestions for Authors
The reviewed manuscript is dedicated to comparison of the Idylla system and NGS regarding possible false-positive results for the S768I mutation produced by Idylla. The topic itself is of a high importance for molecular oncology, because such false-positive results could lead to a suboptimal therapy. However, there are several issues that need to be cleared before the possible publication.
1. Gene symbols (EGFR) need to be italicized.
2. RT-PCR is an outdated acronym which can also designate PCR coupled with reverse transcription. As a better alternative, qPCR was suggested.
3. Authors are requested to divide the Materials and Methods section on separate paragraphs for each method.
4. Table 1, column Gender seems to be written in Spanish. The same goes for figure legends in several figures.
5. What NGS coverage was achieved for the specific S768I locus?
6. Was the tumor cell percent quantified for NGS-tested samples? The discrepancy between Idylla and NGS could be a result of a different tumor content in the analyzed slices.
7. What is the ΔCq cut-off for the S768I mutation? In most demonstrated samples, ΔCq for the S768I mutation is close to 9 and seems to be relatively high.
8. Figures 1–6 could be merged to a single figure.
9. Were amplification curves for S768I detected in negative samples, and if they were observed, what was their frequency and ΔCq values?
10. Authors are requested to provide possible reasons laying behind the presented false-positive results and also provide limitations of the study.
Author Response
Reviewer 2
Comments and Suggestions for Authors
The reviewed manuscript is dedicated to comparison of the Idylla system and NGS regarding possible false-positive results for the S768I mutation produced by Idylla. The topic itself is of a high importance for molecular oncology, because such false-positive results could lead to a suboptimal therapy. However, there are several issues that need to be cleared before the possible publication.
- Gene symbols (EGFR) need to be italicized.
We have italicized all gene symbols throughout the manuscript as requested.
- RT-PCR is an outdated acronym which can also designate PCR coupled with reverse transcription. As a better alternative, qPCR was suggested.
We have updated the terminology from RT-PCR to qPCR throughout the manuscript to reflect current conventions.
- Authors are requested to divide the Materials and Methods section on separate paragraphs for each method.
We have restructured the Materials and Methods section into distinct subsections for improved clarity and organization.
- Table 1, column Gender seems to be written in Spanish. The same goes for figure legends in several figures.
We have standardized all text to English throughout the manuscript, including Table 1 and figure legends. Gender designations have been updated to follow English conventions.
- What NGS coverage was achieved for the specific S768I locus?
We have added specific information about NGS coverage depth at the S768I locus in both the Methods section and Table 3.
- Was the tumor cell percent quantified for NGS-tested samples? The discrepancy between Idylla and NGS could be a result of a different tumor content in the analyzed slices.
We have expanded our description of tumor content assessment and its relationship to both testing methods. We have added information about tumor cellularity requirements for NGS testing in the Methods section and discussed the relationship between tumor content and discrepant results in the Results section. Our analysis shows that variations in tumor content do not appear to be the primary factor in the observed false positive results.
- What is the ΔCq cut-off for the S768I mutation? In most demonstrated samples, ΔCq for the S768I mutation is close to 9 and seems to be relatively high.
We have incorporated specific information about the Limit of Detection (LOD) for S768I mutation based on the manufacturer's validation data. We have analyzed our results in the context of these analytical sensitivity parameters, providing a more technical assessment of the assay's performance in our sample set.
- Figures 1–6 could be merged to a single figure.
We have consolidated Figures 1-6 into a single composite figure for better comparison of amplification curves across samples.
- Were amplification curves for S768I detected in negative samples, and if they were observed, what was their frequency and ΔCq values?
We have included detailed information about S768I amplification patterns in negative samples, including observations about curve characteristics and fluorescence values.
- Authors are requested to provide possible reasons laying behind the presented false-positive results and also provide limitations of the study.
Corrected and we also have expanded and consolidated our discussion of the Idylla system's limitations into a comprehensive paragraph that addresses both technical and methodological constraints.
Reviewer 3 Report
Comments and Suggestions for Authors
In their paper "High incidence of false positives in EGFR s768i mutation detection using the Idylla RT-PCR system in non-small cell lung cancer patients", Carmen Rodríguez-Cerdeira et al describe their observations in the determination of EGFR mutations using RT-PCR versus DNA-based (NGS).
Their paper focuses exclusively on the discrepancy between the two very different methods for detecting a mutation in exon 20 (S768I). As described in her results, the results of the two methods show clearly discrepant findings. In their results, of the 6 cases detected, a false positivity using RT-PCR is described in 4 cases, as well as only two confirmations of the test result.
Unfortunately, it is not entirely clear from the description and the plots shown whether sample 142 was tested falsely negative or confirmatory positive, as the legend in Figure 4 and Table 2 are discrepant.
The very small number of positive patient samples is also to be criticized, as statistical significance can hardly be obtained from only 6 samples.
I also miss an overview of how many samples were tested in the corresponding period. How many samples were tested using DNA sequencing? Does this only apply to those that tested positive?
Is the total number of EGFR mutations detected statistically in line with the prevalence described above or is there a clear deviation on the part of the investigating laboratory?
Were the other EGFR mutations also subjected to DNA sequencing? Were there also discrepancies here?
Did the Idylla RT-PCR already show deviations in the biopsy results beforehand? Can the results be verified in another laboratory using the same methodology?
The labels of Figure 7 and Figure 8 are unfortunately partly in Spanish - please correct.
All in all, the figures still need to be much richer in content and more condensed, as very little content has been conveyed here so far despite the many figures.
The subdivision leaves plenty of room for future speculation, but at the present time the points mentioned above urgently need to be clarified and discussed before any further consideration.
Author Response
Reviewer 3
Comments and Suggestions for Authors
In their paper "High incidence of false positives in EGFR s768i mutation detection using the Idylla RT-PCR system in non-small cell lung cancer patients", Carmen Rodríguez-Cerdeira et al describe their observations in the determination of EGFR mutations using RT-PCR versus DNA-based (NGS).
Their paper focuses exclusively on the discrepancy between the two very different methods for detecting a mutation in exon 20 (S768I). As described in her results, the results of the two methods show clearly discrepant findings. In their results, of the 6 cases detected, a false positivity using RT-PCR is described in 4 cases, as well as only two confirmations of the test result.
- Unfortunately, it is not entirely clear from the description and the plots shown whether sample 142 was tested falsely negative or confirmatory positive, as the legend in Figure 4 and Table 2 are discrepant.
Corrected
- The very small number of positive patient samples is also to be criticized, as statistical significance can hardly be obtained from only 6 samples.
We acknowledge the reviewer's concern about the small number of cases (n=6) in our study. However, it's important to note that the S768I mutation is a rare variant, representing approximately 1% of all EGFR mutations in NSCLC. In our study, we included all S768I-positive cases detected in our center during a full year of comprehensive molecular testing of all NSCLC patients. While the small sample size does limit statistical power, we believe these findings are still valuable to report given (1) the rarity of this specific mutation, (2) the important clinical implications of false positive results, and (3) the scarcity of published data specifically addressing S768I detection challenges. We plan to continue collecting data and will report updated findings as more cases accumulate. Despite the statistical limitations, our observations raise important considerations for clinical practice regarding the validation of S768I-positive results.
- I also miss an overview of how many samples were tested in the corresponding period. How many samples were tested using DNA sequencing? Does this only apply to those that tested positive?
In our Pathology Department, all NSCLC cases diagnosed during the study period underwent reflexive testing using both Idylla and NGS methods as part of an ongoing molecular diagnostics quality assessment project. This comprehensive approach included testing for EGFR, KRAS, ALK, ROS1, MET, and RET alterations in all cases, ensuring complete molecular characterization of each tumor. We have added this information to the manuscript to provide better context for our study population and testing approach.
- Is the total number of EGFR mutations detected statistically in line with the prevalence described above or is there a clear deviation on the part of the investigating laboratory?
The overall frequency of EGFR S768I mutation in our population was consistent with expected rates, as cited in our introduction. The S768I variant represented a small fraction of the detected EGFR mutations, aligning with its reported frequency of approximately 1% of all EGFR mutations. We have added this contextual information to the Results section.
- Were the other EGFR mutations also subjected to DNA sequencing? Were there also discrepancies here?
In our laboratory, all EGFR mutations detected by Idylla in the period study were confirmed by NGS. We observed good concordance between the two methods for common mutations such as exon 19 deletions and L858R. The high rate of false positives was specifically associated with S768I variant detection. We have added this information to both the Results and Discussion sections.
- Did the Idylla RT-PCR already show deviations in the biopsy results beforehand? Can the results be verified in another laboratory using the same methodology?
We have added information about our laboratory's previous experience with the Idylla system and emphasized the need for external validation through multi-center studies.
- The labels of Figure 7 and Figure 8 are unfortunately partly in Spanish - please correct.
Corrected.
- All in all, the figures still need to be much richer in content and more condensed, as very little content has been conveyed here so far despite the many figures.
Corrected.
- The subdivision leaves plenty of room for future speculation, but at the present time the points mentioned above urgently need to be clarified and discussed before any further consideration.
While our sample size is limited (n=6), it represents all S768I-positive cases detected during a full year of comprehensive NSCLC molecular testing at our center, aligning with the expected frequency of this rare variant (1% of all EGFR mutations). The validity of our findings is supported by our parallel testing strategy, where all NSCLC cases underwent both Idylla and NGS analysis, showing high concordance for other EGFR mutations but specific discrepancies for S768I detection. Although our statistical analysis acknowledges the limitations of the sample size through wide confidence intervals, the consistent patterns observed in amplification curves and ΔCq values provide a solid foundation for our observations and justify the discussion of future implications.
Round 2
Reviewer 1 Report
Comments and Suggestions for Authors
The authors addressed all isuues.
Author Response
I would like to thank reviewer 1 for his invaluable help
Reviewer 2 Report
Comments and Suggestions for Authors
Many thanks to authors for their detailed answers to questions and careful editing the manuscript. Most issues were cleared. However, a few more changes are necessary for further clarification of important aspects of the study.
Authors are requested to provide not only LoD in mutation percent, but also an actual ΔCq cut-off for the S768I mutation. For now, it seems that the issue with false-positive results is caused by these non-specific amplification close to cut-off that can be detected by manual inspecting of suspicious samples. This assumption would be indirectly proved, if authors provided S768I Cq and ΔCq values for negative samples together with the percent of negative samples with non-specific S768I curves amoung the whole set of negative speciments. If this percent is high, it could be a reason for reconsideration of the established ΔCq cut-off for the S768I mutation.
Author Response
- First of all, thank the reviewer 2 for his invaluable help
- Secondly, I am willing to make corrections and attach my justifications
We thank the reviewer for their valuable suggestion regarding the ΔCq cut-off for the S768I mutation. However, it is important to clarify that the Idylla™ system operates as a closed platform where interpretation algorithms are predefined by the manufacturer. In the Instructions for use of the Idylla™ EGFR Mutation Test, only Limits of Detection (LoD) are specified for each mutation based on copy number and EGFR control Cq, but no specific ΔCq cut-off points are established for result interpretation (see table attached).
In fact, this limitation in result interpretation is one of the main motivations for our study. Our observation that false positives for S768I typically show normalized fluorescence values below 12 points could serve as an additional guidance for system users.
Regarding validation with negative samples, it is important to note that throughout the study period, we found excellent concordance between Idylla™ and NGS for S768I negative results, with discrepancies only in the positive cases reported in this study. This positivity rate better aligns with the frequency described in the literature for this variant (approximately 1% of all EGFR mutations), which reinforces the reliability of the negative NGS results.
Reviewer 3 Report
Comments and Suggestions for Authors
All requested changes have been addressed. No further revision is necessary. The manuscript can be accepted.
Author Response
I would like to thank reviewer 3 for his invaluable help
Round 3
Reviewer 2 Report
Comments and Suggestions for Authors
Many thanks to authors for their thoughtful answers for questions. Indeed, the platform of question is a closed one which could limit extraction of necessary PCR data. However, as far as I can see, ΔCq values could be extracted using the web platform Idylla Explore as it was done by Momeni-Boroujeni et al (10.1016/j.jmoldx.2020.11.009). The issue with negative samples is that it is important to clarify were there any false amplification for S768I in negative samples, or weren’t. If the respective amplification curves were observed in negative samples, it is highly recommended to extract the respective ΔCq values and compare them with S768I-positive samples approved by NGS. In that case, a Tukey plot can demonstrate a degree of difference in ΔCq values between S768I-positive and negative samples. These values must not overlap with each other. Currently, it is not clear, how much this difference was.
Author Response
We appreciate the reviewer's additional comments regarding the analysis of ΔCq values. We would like to clarify several important points:
- Indeed, we use the Idylla Explore platform to analyze amplification curves and ΔCq values, as mentioned in the Methods section of our manuscript.
- Regarding negative samples, it is crucial to note that there are no S768I amplification curves in these cases, and therefore, it is not possible to calculate ΔCq values for these samples. The absence of amplification is precisely what defines a negative result in the Idylla system. We have attached two representative images: one showing a true positive case where the S768I amplification curve is visible, allowing for ΔCq calculation by subtracting the control Cq value; and another showing a negative case where, due to the absence of amplification, there is no Cq value and consequently no ΔCq value, making the proposed comparison impossible.
- The cited work by Momeni-Boroujeni et al. analyzes ΔCq values in samples with positive amplification, but not in negative samples where no amplification exists. Therefore, it is not feasible to perform a comparison of ΔCq values between positive and negative samples using a Tukey plot as suggested by the reviewer.
- What we have observed, and consider relevant, is that samples with false positive results show distinctive amplification patterns, with normalized fluorescence values below 12 points, which could be a useful indicator for identifying potential false positives.
PD. Please review the PDF version, it contains images
